# Microscopic Description of Platelet Aggregates Induced by *Escherichia coli* Strains

**DOI:** 10.3390/cells11213495

**Published:** 2022-11-04

**Authors:** Amina Ezzeroug Ezzraimi, Jean-Pierre Baudoin, Antoine Mariotti, Laurence Camoin-Jau

**Affiliations:** 1Aix Marseille University Faculté de Médecine t de Pharmacie, IRD, APHM, MEPHI, IHU Méditerranée Infection, 13385 Marseille, France; 2IHU Méditerranée Infection, Boulevard Jean Moulin, 13385 Marseille, France; 3Laboratoire d’Hématologie, Hôpital de la Timone, APHM, Boulevard Jean-Moulin, 13385 Marseille, France

**Keywords:** platelets, *Escherichia coli*, platelet clumps, platelet activation, platelet aggregation, microscopy

## Abstract

In addition to their role in haemostasis, platelets are also involved in the inflammatory and antimicrobial process. Interactions between pathogens and platelets, mediated by receptors can lead to platelet activation, which may be responsible for a granular secretion process or even aggregation, depending on the bacterial species. Granular secretion releases peptides with bactericidal activity as well as aggregating factors. To our knowledge, these interactions have been poorly studied for *Escherichia coli* (*E. coli*). Few studies have characterised the cellular organization of platelet-*E. coli* aggregates. The objective of our study was to investigate the structure of platelet aggregates induced by different *E. coli* strains as well as the ultrastructure of platelet-*E. coli* mixtures using a scanning and transmission electron microscopy (SEM and TEM) approach. Our results show that the appearance of platelet aggregates is mainly dependent on the strain used. SEM images illustrate the platelet activation and aggregation and their colocalisation with bacteria. Some *E. coli* strains induce platelet activation and aggregation, and the bacteria are trapped in the platelet magma. However, some strains do not induce significant platelet activation and are found in close proximity to the platelets. The structure of the *E. coli* strains might explain the results obtained.

## 1. Introduction

Although platelets have many functions, these fragments of megakaryocytes play an essential role in haemostasis [1,2]. Platelets contain numerous specialised organelles dedicated to various functions related to inflammatory and antimicrobial processes. Platelets can interact with bacteria, which can lead to their activation and aggregation [3,4]. However, these mechanisms depend on several factors, mainly the bacterial species and even the strain studied.

Although a great deal of research has been conducted on interactions between platelets and Gram-positive bacteria [5,6,7], which can interact with platelets indirectly via von Willebrand Factor (vWF) or directly. *Staphyloccocus aureus*, *Streptococcus gordonii* and *Streptococcus sanguinis* bind directly to platelets by involving proteins (SrpA, GrspB and SrpA respectively) via the platelet GPIbα [7]. The data on Gram-negative bacteria, in particular *Escherichia coli*, remain insufficient to understand the molecular mechanism of these interactions and to understand the factor of variability of the results. This interaction has been shown to be primarily dependent on TLR4 binding with LPS or by FcγRII recruitment [8].

*E. coli* is involved in sepsis, especially in the elderly and new-borns. Bacteria act directly on the platelets, which can lead to vascular complications with states of immunothrombosis. Therefore, it is very important to study the interaction between platelets and *E. coli* in order to have more knowledge about this interaction and its consequences.

Recently, we studied the antibacterial effect of platelets on different strains of *E. coli*. It was found that this bactericidal activity was strain dependent [9]. We also observed a correlation between this bactericidal effect and the capacity of strains to induce platelet activation [9]. Indeed, some *E. coli* strains induced platelet activation, a result that brought into question the platelet aggregation capability of these strains [8,9].

Electron microscopy (EM) has been a crucial tool in the study of platelet biology and thrombosis for more than 70 years [10,11,12,13]. We previously used scanning electron microscopy (SEM) to characterise the platelet-bacteria aggregates for different bacteria species, namely *Staphylococcus aureus*, *Enterococcus faecalis* and *Streptococcus sanguinis*. [6]. The aim of this study was to describe, with a large panel of SEM and transmission electron microscopy (TEM) techniques, the morphology and ultrastructure of platelet aggregates induced by three different strains of *E. coli*.

## 2. Materials and Methods

### 2.1. Preparation of the Washed Platelets

Blood was drawn by venepuncture in sodium citrate from healthy subjects who were not receiving antibiotic, anti-inflammatory, or anti-platelet drugs. Platelet rich plasma (PRP) was prepared according to the guidelines of the International Society on Thrombosis and Haemostasis (ISTH) [14]. A platelet count was performed using a haematology analyser. The PRP was again centrifuged at 1100× *g* for ten minutes to obtain a platelet pellet that was suspended in phosphate buffered saline (PBS) to obtain a solution of 4 × 10^9^/L. The platelets were then kept at 37 °C in order to prevent activation. The protocol was approved by the ethics committee of the IHU Méditerranée Infection (Reference 2016–002). All subjects gave their written informed consent in accordance with the Declaration of Helsinki.

### 2.2. Preparation of Bacteria

The strains used in this study were selected from each group based on our previous results [9]. The two main selection criteria are the capacity of the strain to induce platelet activation and the profile toward the platelet inhibitory effect (Table 1).

The strains represent the following profiles: *E. coli* J53, platelet sensitive strain which induces platelet activation; *E. coli* K12, platelet resistant strain which induces platelet activation; and *E. coli* LH30, platelet resistant strain which does not induce platelet activation (Table 1).

Identification was confirmed using matrix-assisted laser desorption/ionization time-of-flight (MALDI-TOF) mass spectrometry and the Biotyper database (Bruker, Dresden, Germany). Strains were grown at 37 °C in an overnight culture of Columbia agar +5% sheep blood (bioMérieux, Marcy l’Etoile, France). After 18 h of incubation at 37 °C, the colonies were removed and suspended in 0.9% NaCl medium to obtain the required concentrations: 1 × 10^8^ CFU (colony format units).

### 2.3. Scanning Electron Microscopy (SEM) of Whole Platelet-Bacteria Aggregates

As previously described [6], 200 µL of living PBS-washed platelets (4 × 10^8^/mL) and of PBS-washed bacteria (10^9^ CFU/mL) were mixed for one hour at 37 °C [9], under rotation to avoid the static state and the development of aggregates due to gravity. Cells were then fixed using 2.5% glutaraldehyde in 0.1 M sodium cacodylate buffer for one hour. After fixation, samples were rinsed three times with 0.1 M sodium cacodylate (five minutes each) to remove any residual fixative. Cells were dehydrated with graded ethanol concentrations: 25% for five minutes; 50% for five minutes; 70% for five minutes; 85% for five minutes; 95% for five minutes (twice); 100% ethanol for 10 min (three times). Following ethanol dehydration, cells were incubated for five minutes in an ethanol/hexamethyldisilazane (HMDS, Sigma Aldrich, USA) (1:2) mixture, then twice in pure HMDS. Between all steps, cells were gently stirred and centrifuged at 1300 rpm. A drop of cells in pure HDMS was deposited on a glass slide and allowed to air dry for 30 min before observation [6,15] Cells were visualised with a TM4000Plus (Hitachi, Tokyo, Japan) scanning electron microscope operated at 10 kV with Back-Scattered Electrons (BSE) detector at magnifications ranging from X200 to X3000.

### 2.4. SEM of Ultra-Thin Sections of Platelet-Bacteria Aggregates

Cells mixtures were fixed with glutaraldehyde (2.5%) in 0.1 M sodium cacodylate buffer. Resin embedding was microwave-assisted with a PELCO BiowavePro+. Samples were washed with a mixture of 0.2 M saccharose/0.1 M sodium cacodylate and post-fixed with 1% OsO4 diluted in 0.2 M potassium hexa-cyanoferrate (III)/0.1 M sodium cacodylate buffer. After being washed with distilled water, samples were gradually dehydrated by successive baths containing 30% to 100% ethanol. Substitution with Epon resin was achieved by incubations with 25% to 100% Epon resin. Resin was heat-cured for 72 h at 60 °C. Ultrathin 100 nm sections were cut and placed on HR25 300 Mesh Copper/Rhodium grids (TAAB). Sections were contrasted with uranyl acetate and lead citrate according to Reynolds’s method [16]. Grids were attached with double-side tape to a glass slide and platinum-coated at 10 mA for 20 s with a MC1000 sputter coater (Hitachi High-Technologies, Japan). Electron micrographs were obtained on a SU5000 scanning electron microscope (Hitachi High-Technologies, Japan) operated in high-vacuum at 7 kV accelerating voltage and observation mode (spot size 30) with BSE detector. The abundance of bacteria in each mixture was determined by measuring the surface occupied by the bacteria using Fiji software with 10 images for each mixture. The criteria used to indicate that the platelets are hyper-activated are: the presence of pseudopodia and the presence of platelet aggregates; moderately activated: the presence of pseudopodia and the absence of platelet aggregates; not activated: presence of intact platelets in lenticular shape, absence of pseudopodia and platelet aggregates. The aggregates presence was determined by measuring the surface occupied by the platelet aggregates using Fiji software with 10 images for each mixture.

### 2.5. Transmission Electron Microscopy (TEM) of Negatively Stained Bacteria

Samples of pure bacteria were fixed with glutaraldehyde (2.5%) in 0.1 M sodium cacodylate buffer. A drop of fixed bacterial suspension was applied for five minutes to the top of a formvar carbon 400 mesh nickel grid (FCF400-Ni, EMS), which was previously glow discharged. After drying on filter paper, bacteria were immediately stained with aqueous 1% ammonium molybdate (1-800- ACROS, USA) for 10 s. After drying, electron micrographs of negatively stained bacteria were acquired using a Tecnai G2 transmission electron microscope (Thermo-Fischer/FEI) operated at 200 keV equipped with a 4096 × 4096 pixels resolution Eagle camera (FEI).

### 2.6. SEM of Whole Bacteria

Bacteria were fixed with 2.5% glutaraldehyde in 0.1 M sodium cacodylate buffer for at least one hour. After fixation, bacteria were rinsed for one minute with 0.1 M sodium cacodylate. Bacteria were gradually dehydrated with increasing ethanol concentrations: 30%, 50%, 70%, 90%, and 100% (one minute each). Bacteria were incubated for one minute in ethanol/hexamethyldisilazane (HMDS, Sigma Aldrich, USA) with a 1:2 ratio and finally incubated in pure HMDS. Between all previous steps, cells were gently stirred and centrifuged at 5000 rpm. Finally, 100 µL of each bacteria solution was centrifuged on a cytospin glass slide at 800 rpm for eight minutes. After deposition, bacteria were air-dried for five minutes and slides were platinum sputter-coated for 20 s at 10 mA (Hitachi MC1000). Observations were made using a SU5000 (Hitachi High-Technologies, Tokyo, Japan) SEM with Secondary-Electrons (SE) detector in high-vacuum mode at 1 kV acceleration voltage, observation mode (spot size 30).

### 2.7. Statistical Analysis

Statistical analyses were performed using GraphPad Prism 9 (86) for Mac OS X (San Diego, CA, USA, www.graphpad.com, accessed on 27 October 2022) Significant differences (for occupied surface and reliefs thickness) between two groups were determined using the two-tailed, paired Student’s *t*-test. The statistical significance was set at *p* < 0.05.

## 3. Results

### 3.1. SEM of Whole Platelet-Bacteria Mixtures

To describe the spatial relationship between platelets and *E. coli* after mixing, we used BSE-SEM, detecting back-scattered electrons. Analysis was performed on washed platelets from healthy subjects incubated with three strains of *E. coli.*

BSE-SEM of whole platelet-bacteria mixtures showed that in the case of the K12-platelets aggregates (Figure 1A1,A2), platelets were moderately activated and there were a few platelet aggregates, with many bacteria above these aggregates. In LH30-platelets mixtures (Figure 1B1,B2D), there were many bacteria but few platelet aggregates. In contrast, the J53 strain induced a higher activation as seen by the morphology of the few non-aggregated platelets, and a greater aggregation. Many platelet clumps were found but surprisingly, almost no bacteria were detected in this mixture of platelets-J53 (Figure 1C1,C2). The abundance of bacteria was determined by measuring the occupied surfaces by the bacteria in each platelet-bacteria preparation (Figure 2A). The bacteria present in the platelet-J53 preparation were the least abundant compared to the 2 other preparations (significant differences: *p* < 0.0001 between J35 and K12, *p* = 0.0029 between J53 and LH30). Measure of the occupied surfaces by the platelet aggregates in each platelet-bacteria preparation (Figure 2B) have shown a significantly higher surface area in the platelet –J53 preparation compared to the two other preparations (Figure 2B). The results of this part have been summarized in Table 2. 

To check whether bacteria were indeed absent from the platelet aggregates after mixing, we next performed resin-embedding and ultra-thin sectioning on platelets-J53 mixtures to access the internal content of the aggregates.

### 3.2. SEM of Ultra-Thin Sections of Platelet-Bacteria Mixtures

We performed SEM of the ultra-thin sections of the resin-embedded platelets-bacteria aggregates, using the back-scattered electron (BSE) detector, and these images showed that there were differences in the ultrastructural organisation of the different mixtures regarding the strains.

Indeed, platelets and K12 strain *E. coli* bacteria (Figure 3A1,A2) were found to be mixed within the aggregates. Platelets were intact and moderately activated, with no release of granular content. The bacteria were found in significant numbers between the pseudopods of the activated platelets. For LH30, platelets and bacteria were found side-by-side rather than mixed together (Figure 3B1,B2). Although platelets were found intact, as with K12, platelets were not as activated as with the K12 strain, with a few pseudopods and intact granular contents. In contrast, in the case of the J53 strain, the granular content of the platelets was observed to be released extracellularly into an amorphous matrix (Figure 3C1,C2). *E. coli* J53 bacteria were found to be trapped inside this matrix. These latter results explain the absence of J53 bacteria on the surface of the whole platelet aggregates observed by SEM (Figure 1). The results of this part have been summarized in Table 2. 

To highlight the morphology of platelet-*E. coli* aggregates, we designed this table to be able to compare the aggregates according to the strains, using the two techniques. In order to describe these aggregates, we chose the following criteria: the visibility of platelets and bacteria, the aspect of platelets, platelet activation, the aspect of platelet granules and the colocalisation of bacteria and platelets. NC: non conclusive.

### 3.3. Electron Microscopy of Bacteria

In order to understand whether ultrastructural differences between the three *E. coli* bacteria strains could explain their respective behaviour regarding platelet aggregation, we performed an in-depth electron microscopy analysis on the cellular level.

First, we analysed the ultrastructure of the bacteria in ultra-thin sections of platelets-bacteria aggregates (Figure 4). We found that for K12 bacteria, the cell wall was regular and attached to the periplasm, and that bacteria possessed electron-dense bodies within an electron-lucent periplasm (Figure 4A1,A2). For LH30 and J53 bacteria, we found close ultrastructures, with irregular or sinuous cell walls for LH30 and J53 bacteria, respectively (Figure 4B1,B2,C1,C2). LH30 and J53 cell walls were found to be detached from an electron-dense periplasm, with a more pronounced detachment for LH30 (Figure 4B2,C2).

Secondly, we analysed whole bacteria by TEM and SEM. When negatively stained and imaged by TEM, the three *E coli* bacteria strains had an elongated shape and J53 strain presented flagella (Figure 5A1,B1,C1). To better describe the surface of the cells, we next performed SEM of the whole bacteria using the secondary electrons (SE) detector. The results of this part have been summarized in Table 3.

Using SE-SEM, we observed that K12 and J53 *E. coli* bacteria possessed a ‘rough’ surface, composed of a complex network of many thin surface reliefs (Figure 5A2,C2; 14 cells analysed). Measurement of membrane thickening was performed using Fiji software with X bacteria analysed for each strain. In contrast, the surface of LH30 *E coli* bacteria was ‘smooth’, with a simpler network of larger surface reliefs (Figure 4B2; 14 cells analysed). The thickness of these bacterial reliefs surface was measured, and the results were analysed statistically. The analysis showed that indeed the J53 strain has reliefs that are less thick (47.34 µm ± 18.23) compared to the two other strains (107.8 µm ± 26.81 and 115.2 µm ± 43.04 for K12 and LH30 respectively. *p* = 0.001 and *p* = 0.0008 for K12 and LH30 respectively. No significant difference was observed between K12 and LH30 strains). These SEM images also confirmed the presence of flagella on the J53 strain *E. coli* bacteria (not shown) and were used for measuring bacteria dimensions. The average dimensions of the bacteria were 2854 ± 961 nm length and 826 ± 71 nm width for K12 (n = 14), 2570 ± 731 nm length and 1073 ± 121 nm width for LH30 (n = 14) and 2931 ± 483 nm length and 917 ± 329 nm width for J53 (n = 14). No significant differences were observed between strains regarding length and width. The results of this part have been summarized in Table 3.

## 4. Discussion

This study described the consequences of the interaction between platelets and three strains of *Escherichia coli* using complementary electron microscopy techniques. Our overall results show that the appearance of aggregates and the colocalisation of bacteria and platelets are strain-dependent. Structural analysis by electron microscopy of the strains could explain our results. This structural variability of the platelet-*E. coli* mixture confirmed our previous results and complements them [9].

Few studies have used fluorescence microscopy to describe the colocalisation of platelets and *E. coli* [12,13]. To our knowledge, none have characterised platelet-*E. coli* mixtures by electron microscopy. The SEM study allowed the analysis of the activation state of the platelets and in particular the granular secretion process. Three platelet activation profiles were obtained depending on the strain tested. Platelet activation would, therefore, be strain dependent, as previously demonstrated by Watson on a reduced panel of two strains [17]. The study of platelet-LH30 mixtures shows the persistence of intra-platelet granules, demonstrating the absence of an activating effect of this strain on platelets, in agreement with the flow cytometry results obtained previously [9]. LH30 is a clinical and colistin-resistant isolate. However, this profile did not allow us to explain the observed patterns. We demonstrated in previous work that *E. coli*-induced platelet activation is independent of the response to colistin [9]. The K12 strain induced moderate activation, as previously observed by Fejes et al. and by our team [9,18]. In contrast, a strong activation, responsible for the formation of an aggregate, was detected for the J53 mixtures. This is both surprising and interesting, because the J53 strain is a mutant of K12.

Among the analytical criteria of our study, we were interested in the number of bacteria present in the mixtures. Comparative SEM analysis of the three platelet-bacteria mixtures shows that the abundancy of bacteria present on the platelet-*E. coli* mixtures is variable depending on the strain tested. It can be hypothesised that the number of bacteria found is a consequence of the bactericidal activity of the platelets. Indeed, platelet activation leads to the release of granular content and, in particular, the release of platelet microbicidal peptides (PMPs), which have bactericidal activity on certain strains. We have previously shown that there is an inverse correlation between the activation state induced by the strains tested and their bactericidal power [9]. This explains why high bacterial abundancy was observed with strain LH30, which did not induce platelet activation, as shown by the persistence of intra-platelet granules. In contrast, in the presence of strain J53, which was responsible for strong aggregation, no bacteria were observed in the SEM images. Whole mount analysis allowed us to detect bacteria trapped in the platelet magma and in low abundancy.

In order to understand the difference in behaviour of the two strains, we analysed the ultrastructure of the bacteria when mixed with platelets in ultra-thin sections by BSE-SEM. The presence of electron-dense bodies in the periplasm of strain K12 was detected which, according to the literature, may be deposits of polyphosphates formed as a result of a defect in LPS synthesis [19]. These images may illustrate that the K12 strain has a deletion of a mobile IS5 element in the WbbL gene involved in O-antigen biosynthesis [20], meaning that the two strains do not have the same form of LPS. Our hypothesis is that the contrasting behaviour of the two strains toward platelets could be linked to the difference in the LPS form, more precisely antigen-O. Furthermore, this hypothesis is also supported by the fact that it has been shown that platelet TLR-4 is involved in the recognition of LPS [4,21,22]. This TLR-4 signalling pathway is capable of inducing platelet aggregation [23,24]. Our results would help to understand the difference in behaviour between the K12 and J53 strains.

Another point that caught our attention and that we feel should be highlighted is the morphology of bacteria. In fact, the K12 and LH30 strains have a kind of network on their surfaces which is mainly characterised by reliefs. A complex network with thin reliefs was observed on the K12 strain surface, while the LH30 strain has a less complex network on its surface, but with thicker reliefs. These characteristics lead us to suggest that this surface network could be a barrier against the interaction with platelets that prevents activation, and also prevents the effect of PMPs in the case of activated platelets. This difference in structure may explain the difference in behaviour between these two strains.

Our results allow us to better understand our previous work and enable us to understand the heterogeneity of the response of *E. coli* strains against platelets by combining several electron microscopy techniques.

According to our data, the bactericidal mechanism would be of secretory origin. Different scenarios can be distinguished. The platelets are not activated by the bacteria and do not inhibit bacterial growth, which is the case for strain LH30. The absence of platelet activation, probably related to the membrane structure of the strain, which constitutes a physical barrier with the platelets, does not induce the release of PMPs of granular origin. The moderate platelet activation induced by strain K12 is insufficient to induce secretion of granular content. We consider these strains to be platelet resistant. Conversely, the J53 strain strongly activates platelets, responsible for significant platelet aggregation. The bacteria are then trapped in the platelet aggregates and their growth is inhibited. This strain is considered platelet-sensitive. The difference between the behavior of J53 and K12 strains towards platelets is probably related to the absence of a functional O-antigen, which is then reflected in the absence of degranulation”.

Since laboratory strains generally behave differently from clinical or wild-type strains, we will need to confirm our observations on a larger number of clinical strains.

The different strain profiles might have important clinical consequences in patients. We could characterize the functional profile of strains in *Escherichia coli* bacteremia by platelet aggregation methods or by ex silico serotyping. Analysis of these data in a prospective study in patients with *Escherichia coli* bacteremia could allow us to validate the clinical relevance of these data.

## Figures and Tables

**Figure 1 cells-11-03495-f001:**
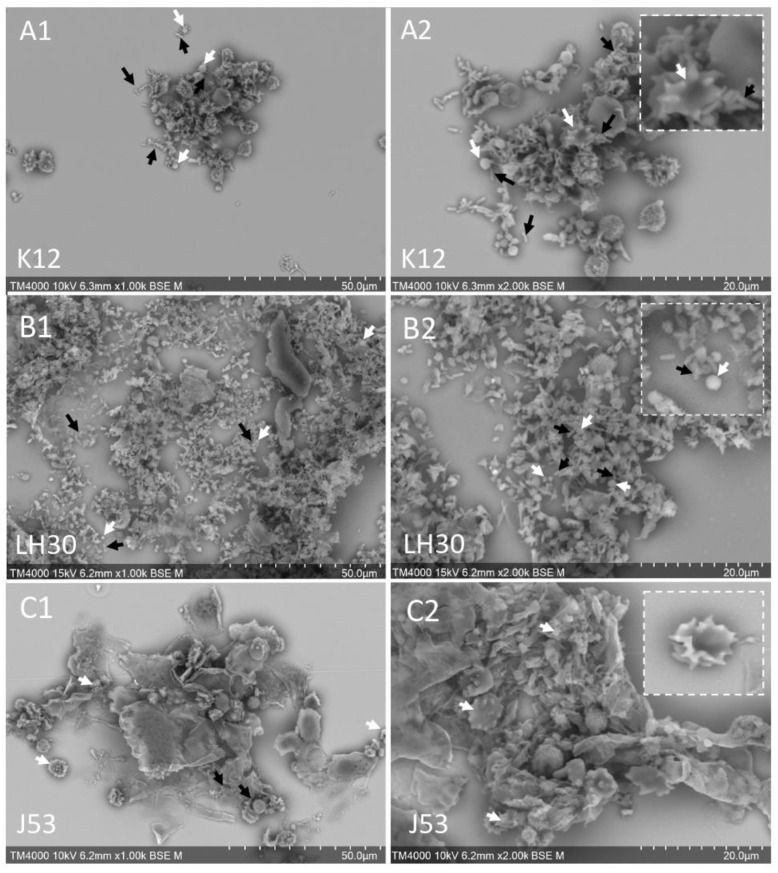
BSE-SEM observation of whole platelets-*E. coli* mixtures. (**A1**,**A2**): Platelets and *E. coli* K12 mixture; (**B1**,**B2**): Platelets and *E. coli* LH30 mixture; (**C1**,**C2**): Platelets and *E. coli* J53 mixture. White arrows: platelets, black arrows: bacteria.

**Figure 2 cells-11-03495-f002:**
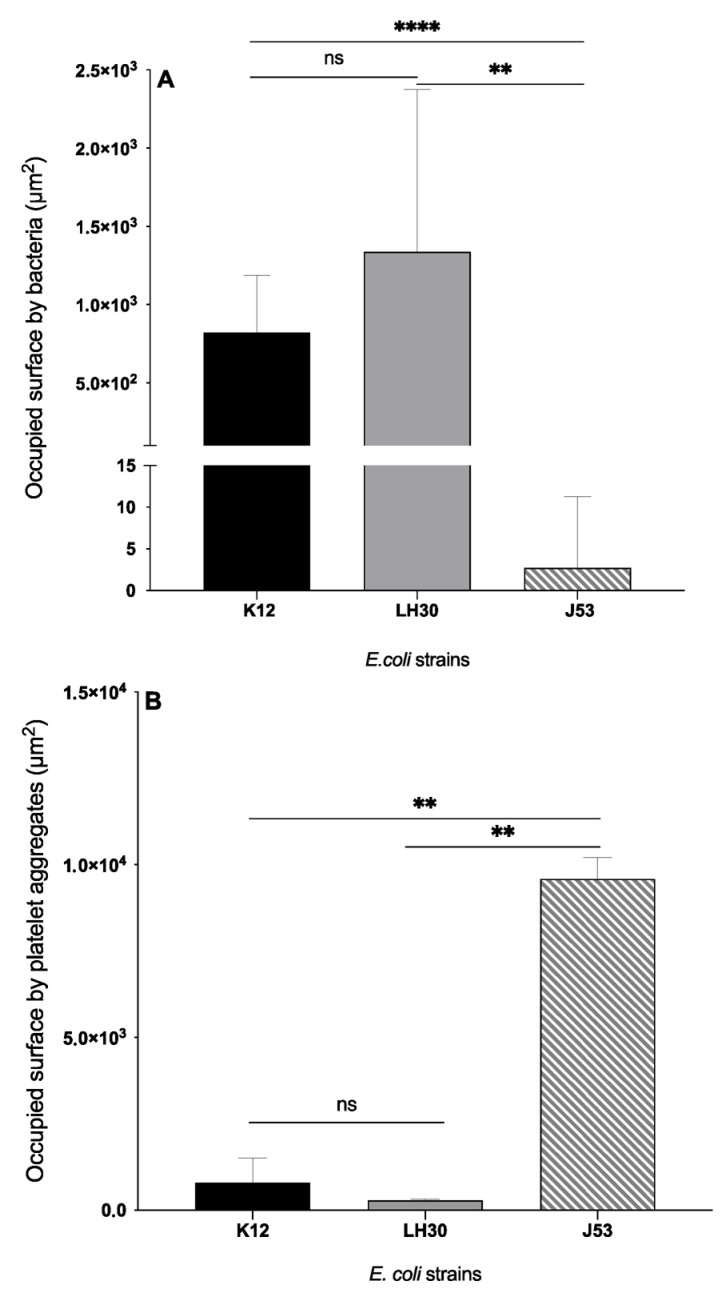
Occupied surface (µm^2^) by bacteria (**A**) and platelet aggregates (**B**) for each *E. coli* strain in platelet-bacteria mixtures. (**A**): Bars represent Mean with SD. Black column: occupied surface by *E. coli* K12 (8.23 × 10^2^ ± 3.64 × 10^2^). Grey column: occupied surface by *E. coli* LH30 (1.34 × 10^3^ ± 1.04 × 10^3^). Striped grey column: occupied surface by *E. coli* J53 (2.71 ± 8.57). (**B**): Bars represent Mean with SD. Black column: occupied surface by platelet aggregates in platelets-K12 mixture (795.6 ± 710.5). Grey column: occupied surface by platelet aggregates in platelets-LH30 mixture (281.2 ± 36.71). Striped grey column: occupied surface by platelet aggregates in platelets-J53 mixture (9583 ± 630.7). **,****: significant difference. ns: non-significant. Significant differences between the two groups were determined using the two-tailed, paired Student. ** *p* < 0.01, **** *p* < 0.0001.

**Figure 3 cells-11-03495-f003:**
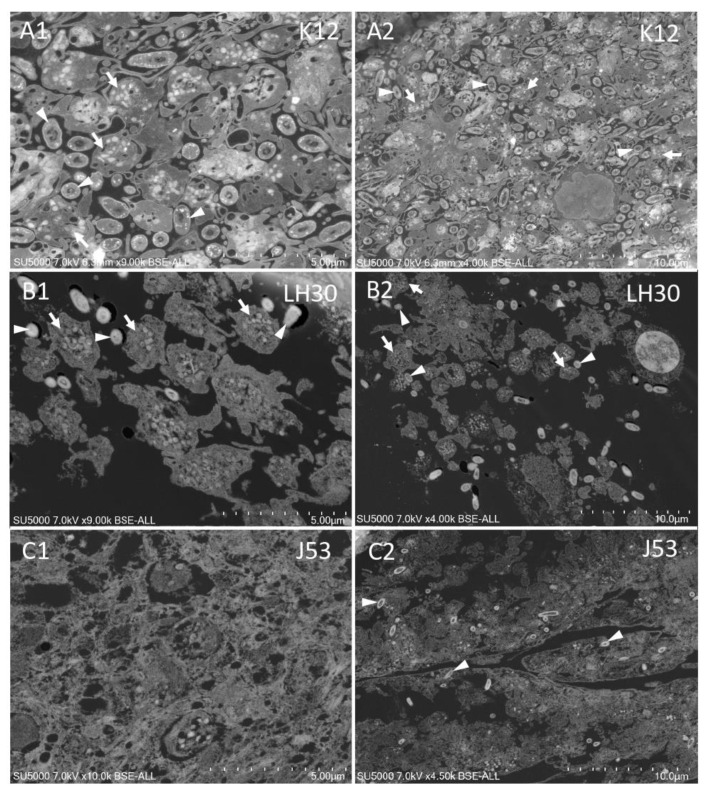
BSE-SEM observation of ultra-thin sections of platelets-*E. coli* mixtures. (**A1**,**A2**): Platelets and *E. coli* K12 mixture, (**B1**,**B2**): Platelets and *E. coli* LH30 mixture; (**C1**,**C2**): Platelets and *E. coli* J53 mixture. White arrows: platelets, white arrowheads: bacteria.

**Figure 4 cells-11-03495-f004:**
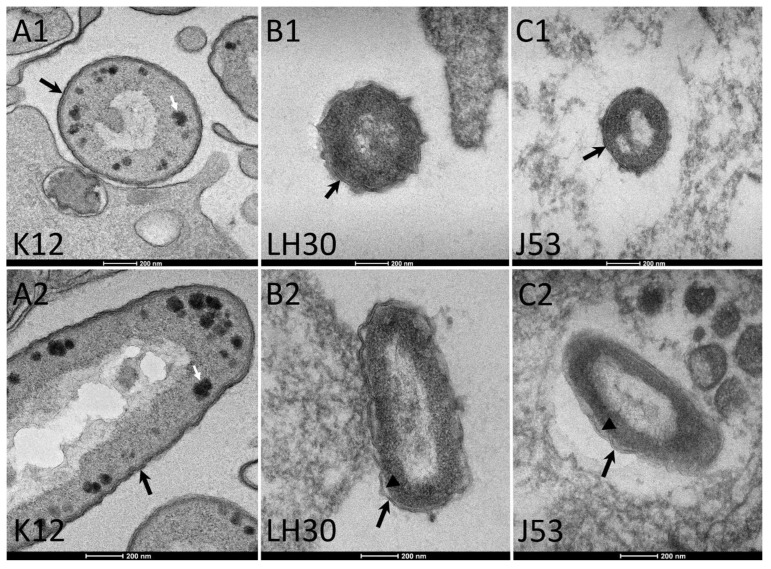
BSE-SEM observation of ultrathin sections of bacteria within the platelet aggregates. (**A1**,**A2**): K12 *E. coli* transversal and longitudinal sections, respectively. (**B1**,**B2**): LH30 *E. coli* transversal and longitudinal sections, respectively. (**C1**,**C2**): J53 *E. coli* transversal and longitudinal sections, respectively. Black arrows: cell wall; white arrows: electron dense bodies; arrowheads: free space between cell wall and periplasm.

**Figure 5 cells-11-03495-f005:**
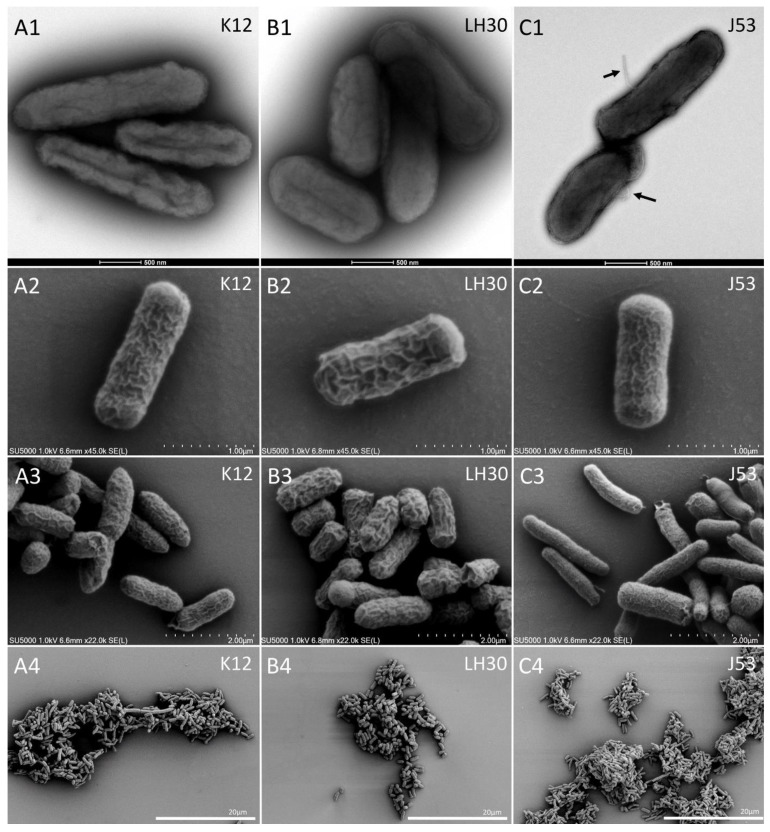
TEM and SE-SEM observation of the whole bacteria strains. (**A1**,**B1**,**C1**): TEM negative staining of K12, LH30 and J53 *E. coli* bacteria, respectively. Arrows in C1 point to flagella. (**A2**–**A4**): Secondary electrons (SE)-SEM of whole K12 *E. coli* bacteria. (**B2**–**B4**): SE-SEM of whole LH30 *E. coli* bacteria. (**C2**–**C4**): SE-SEM of whole J53 *E. coli* bacteria.

**Table 1 cells-11-03495-t001:** Origins and profiles of tested strains.

Strain	Origin	Platelet Activation	Platelet Bactericidal Effect	O-Antigen Serotyping	Reference
K12	Laboratory strain	+	-	-	[9]
LH30	Clinical isolate	-	-	O8	[9]
J53	Laboratory strain	+	+	O16	[9]

Platelet activation: the capacity of the strain to induce platelet activation; (+): induce platelet activation, (-): do not induce. Platelet bactericidal effect: the capacity of platelets to inhibit bacterial growth; (+): growth inhibition, (-): no growth inhibition.

**Table 2 cells-11-03495-t002:** Description of whole and sectioned platelets-bacteria aggregates using SEM.

	BSE-SEM of Whole Platelets-Bacteria Aggregates (Figure 1)			BSE-SEM of Ultrathin Sections of Platelets-Bacteria Aggregates (Figure 2)			
Criteria	Visible Platelets	Visible Bacteria	Platelet Activation	Platelet Integrity	Platelet Activation	Platelet Granules	Bacteria’s Location Regarding Platelets
K12 strain	Yes	Yes	Moderate	Yes	Moderate	Inside platelets	Mixed
LH30 strain	Yes	Yes	No	Yes	No	Inside platelets	Side by side
J53 strain	Moderate	No	NC	Amorphous matrix	High	Among the amorphous matrix	Inside the amorphous matrix

**Table 3 cells-11-03495-t003:** Bacteria morphology.

*E. coli* Strain		BSE-SEM Ultrathin Sections (Figure 3)	TEM Negative Staining (Figure 4)	SE-SEM (Figure 4)
	Technique			
K12		Regular cell wall, attached to periplasm, electron-dense bodies	Elongated shape	Thin surface reliefs
LH30		Irregular shaped cell wall, detached from periplasm, electron-dense periplasm	Elongated shape	Thick surface reliefs
J53		Sinuous cell wall, detached from periplasm, electron-dense periplasm	Elongated shape ± flagella	Thin surface reliefs

## Data Availability

The data presented in this study are available on request from the corresponding author.

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
