# Peer review of "Microscopic Description of Platelet Aggregates Induced by Escherichia coli Strains"

_cells, 2022, doi:10.3390/cells11213495_

Round 1

Reviewer 1 Report

The authors investigated the structure of platelet activation and interaction with 3 different E. coli strains (K12; LH30; J53) by scanning and transmission electron microscopy (SEM and TEM). In recent publications of the authors (refs. 8 and 9), the complex interaction of platelets and E.coli was reported, which is the basis of the present paper as summarized in table 1. This earlier work is now complemented by the analysis of platelet E-coli interactions by various electron microscopy techniques. The impressive EM data support the concept that the platelet – E. coli interaction is strain dependent.

Overall, this work is of high quality, especially the SEM/TEM part. There are a few points where the paper can be improved.

 1)    Functional/ medical implications of their new findings should be more precisely discussed and strengthened in the paper.

2) Did the authors observe (in their present or previous studies)  differences between α- and δ-granule secretion?

3) Did the authors measure bactericidal effects of activated platelets? Which bacteria ?

4)  What can be briefly summarized at the molecular level about the 3 different E.coli strains and their platelet interactions? What is the potential clinical relevance ?

5)   Also, the mechanisms of other bacteria (gram +) should be briefly but precisely compared to the bacteria studied here.  

Author Response

Dear reviewer,

We thank you for your remarks which underline and value the work done in electron microscopy.

We have responded to each of your remarks. We hope that these answers correspond to your expectations.

Best regards

Camoin-Jau

Reviewer 2 Report

The paper entitled "Microscopic Description of Platelet Aggregates Induced by Escherichia coli Strains", by Ezzeroug Ezzraimi, Amina et. al., is an original article of an experimental, in-vitro, evaluation of the ultrastructure of bacteria-platelet aggregates. By using two laboratory strains, K12 and J53, and one clinical LH30 Escherichia coli, the authors mixed each bacterial strain with platelets. The mixtures were incubated at 37°C for one hour until they fixed them and prepared samples for Scanning electron microscopy (SEM). Also, they saw the bacteria morphology by Transmission electron microscopy (TEM). They found that the structure of the aggregates, and the activations of platelets, differ between the bacterial strains. They also foundirregular surface membranes on the bacteria that didn't activate the platelets. The paper is an small increment on another paper published by the same author, in which they showed that some E. coli strains can activate platelets and induce bactericidal activity Ezzeroug Ezzraimi A, Hannachi N, Mariotti A, Rolland C, Levasseur A, Baron SA, Rolain JM, Camoin-Jau L. The Antibacterial Effect of Platelets on Escherichia coli Strains. Biomedicines. 2022 Jun 28;10(7):1533. doi: 10.3390/biomedicines10071533. PMID: 35884840. In the current paper they add another detail “platelet aggregation”. 

Commentaries:

  1. It would have been interesting to include more laboratory strains and more wild-type strains in the experiments. This may indicate that wild type E. coli has mechanisms to avoid platelet activation and bactericidal effects. These mechanisms may have been lost in lab strains.
  2. The authors called the "moderately activated", and "not as activated as..." for measuring the platelet activations. In this paper, the authors do not define how they determined the platelet activations. I found this information is a lacking in this paper and the readers need to understand the dimensions of this variable. In their previous paper, the authors defined the platelet activations by their CD62P expression, using flow cytometry. If the authors are using a morphological definition for platelet activations they should mention the features, such as the presence of pseudopods (?), etc. Could it be translated into numerical parameters?
  3. Another problem in the paper is the definition of the aggregate's presence since they use the terms "few platelet aggregates" and "greater aggregation". Please, include or clarify the corresponding definitions for these terms. Could it be translated into numerical parameters?

  1. The authors indicated that the "number" of bacteria, or the "number" of aggregates, as important differences in aggregation. However, they don't define how they visualized the number. The authors need to consider a measure for this finding. For example, since the distribution of bacteria, and aggregates may not be uniform in the slides (grids), they may need to define the number of bacteria and the number of aggregates by the number of fields seen per sample.(2)
  2. Please, include a definition of the terms complex, simple, and less complex network from Table 3. 
  3. Since laboratory strains usually behave differently from clinical or wild-type strains, please, consider addressing this issue as a limitation in the discussion section.

Author Response

Dear reviewer, 

Dear reviewer,

We have responded to each of your remarks. We hope that these answers correspond to your expectations.

This work is effectively a continuation of a recently published work. We do not consider this work as a small increment.  In order to understand these complex bactericidal, activation and aggregation phenomena, we have done important work in electron microscopy. To our knowledge, we are the first to use these different electron microscopy techniques to understand these complex interaction mechanisms.

Best regards

Pr L Camoin-Jau
